# ENHANCED TEMPORAL KNOWLEDGE EMBEDDINGS WITH CONTEXTUALIZED LANGUAGE REPRESENTATIONS

## ABSTRACT

World knowledge exists in both structured (tables, knowledge graphs) and unstructured forms (texts). Recently, there have been extensive research efforts in the integration of structured factual knowledge and unstructured textual knowledge. However, most studies focus on incorporating static factual knowledge into pre-trained language models, while there is less work on enhancing temporal knowledge graph embedding using textual knowledge. Existing integration approaches can not apply to temporal knowledge graphs (tKGs) since they often assume knowledge embedding is time-invariant. In fact, the entity embedding in tKG embedding models usually evolves over time, which poses the challenge of aligning temporally relevant textual information with entities. To this end, we propose **E**nhanced Temporal Knowledge Embeddings with **Co**ntextualized **La**nguage Representations (ECOLA), which uses tKG quadruple as an implicit measure to temporally align textual data and the time-evolving entity representations and uses a novel knowledge-text prediction task to inject textual information into temporal knowledge embedding. ECOLA jointly optimizes the knowledge-text prediction objective and the temporal knowledge embedding objective, and thus, can simultaneously take full advantage of textual and structured knowledge. Since existing datasets do not provide tKGs with aligned textual data, we introduce three new datasets for training and evaluating ECOLA. Experimental results on the temporal knowledge graph completion task show that ECOLA outperforms state-of-the-art tKG embedding models by a large margin.

## 1 INTRODUCTION

Knowledge graphs (KGs) have long been considered an effective and efficient way to store structural knowledge about the world. A knowledge graph consists of a collection of *triples* $(s, p, o)$, where $s$ (subject entity) and $o$ (object entity) correspond to nodes and $p$ (predicate) indicates the edge type (relation) between the two entities. Common knowledge graphs (Toutanova et al., 2015; Dettmers et al., 2018) assume that the relations between entities are static connections. However, in the real world, there are not only static facts and properties but also time-evolving relations associated with the entities. For example, the political relationship between two countries might worsen because of trade fights. To this end, temporal knowledge graphs (tKGs) (Tresp et al., 2015) were introduced that capture temporal aspects of relations by extending a triple to a *quadruple*, which adds a timestamp or time interval to describe when the relation is valid, e.g. (*Argentina*, *deep comprehensive strategic partnership with*, *China*, *2022*). Extensive studies have been focusing on learning temporal knowledge embedding (Leblay & Chekol, 2018; Han et al., 2020c), which not only helps infer missing links in tKGs but also benefits various knowledge-related downstream applications, such as temporal question answering (Saxena et al., 2021b).

However, knowledge graph embedding often suffers from the sparseness of knowledge graphs. For example, the tKG model proposed by Han et al. (2020a) performs much better on the dense tKG than the sparse one. To address this problem, some recent studies incorporate textual information to enrich knowledge embedding. KEPLER (Wang et al., 2021) learns the representation of an entity by encoding the entity description with a pre-trained language model (PLM) and optimizing the knowledge embedding objective. KG-Bert (Yao et al., 2019) takes entity and relation descriptions of

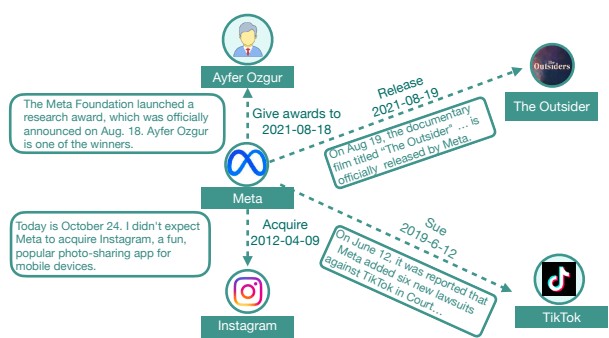

Figure 1: An example of a temporal knowledge graph with textual event descriptions.

a triple as the input of a PLM and turns knowledge graph completion into a sequence classification problem. However, they do not take the temporal nature and the evolutionary dynamics of knowledge graphs into account. In tKG embedding models, the entity representations usually evolve over time as they involve in different events at different timestamps. Taking financial crises as an example, companies are more likely involved in events such as laying off employees. But when the economy recovers, companies hire staff again rather than cut jobs. Thus, the entities should also be able to drift their representations over time to manage the changes. Therefore, given an entity, it should be taken into account which textual knowledge is relevant to it at which timestamp. We name this challenge as *temporal alignment* between texts and tKG, which is to establish a correspondence between textual knowledge and their temporal knowledge graph depiction. This is one of the challenges that existing approaches cannot handle due to their limitation of assuming knowledge embedding is static and using the time-invariant description of an entity to enhance its representation. Thus, they are not appropriate in temporal knowledge graph scenarios where temporal alignment is required. The other challenge is that temporal knowledge embedding models learn the entity representations as a function of time, which exposes another limitation of existing approaches that their architectures cannot be naturally combined with tKG models. Therefore, it is not clear how to enhance *temporal knowledge embedding* with textual data.

To this end, we propose **E**nhanced Temporal Knowledge Embeddings with **Co**ntextualized **La**nguage Representations (ECOLA), which uses temporally relevant textual knowledge to enhance the time-dependent knowledge graph embedding and ensures that the enhanced knowledge embedding preserves the temporal nature. Specifically, we solve the temporal alignment challenge by using tKG quadruples as an implicit measure. We pair a quadruple with its relevant textual data, e.g., event descriptions, which corresponds to the temporal relations between entities at a specific time. Then we use the event description to enhance the representations of entities and predicate involved in the given quadruple. In particular, we encode entities and predicates by tKG embedding models and encode texts using token embedding . Given a quadruple-text pair, we concatenate the embedding of entities, predicate, and textual tokens and feed them into a pre-trained language model. We introduce a novel knowledge-text prediction (KTP) task to inject textual knowledge into temporal knowledge embedding. The KTP task is an extended masked language modeling task, which randomly masks words in texts and entity/predicates in quadruples. With the help of the KTP task, ECOLA would be able to recognize mentions of the subject entity and the object entity and align semantic relationships in the text with the predicate in the quadruple. Thus, the model can take full advantage of the abundant information from the textual data, which is especially helpful for embedding entities and predicates that only appear in a few quadruples. ECOLA jointly optimizes the knowledge-text prediction and temporal knowledge embedding objectives. Since our goal is to develop an approach that can generally improve any potential tKG models, we combine the model with different benchmark tKG embedding models (Goel et al., 2020; Han et al., 2020c; 2021). For training ECOLA, we need datasets with temporal KG quadruples and aligned textual event descriptions, which is unavailable in existing temporal KG benchmark datasets. Thus, we construct three new temporal knowledge graph datasets by adapting two existing datasets, i.e., GDELT (Leetaru & Schrodt, 2013) and Wiki (Dasgupta et al., 2018), and an event extraction dataset (Li et al., 2020). To make a fair comparison with other temporal KG embedding models and keep fast inference, we only take the enhanced

temporal knowledge embedding to perform the temporal KG completion task at test time but do not use any textual descriptions of test quadruples.

To summarize, our contributions are as follows: (i) We propose ECOLA that enhances temporal knowledge graph representation models with textual knowledge via pre-trained language models. ECOLA shows its superiority on the temporal KG completion task and can be potentially combined with any temporal KG embedding model. (ii) We are the first to address the challenge of enhancing temporal knowledge embedding with temporally relevant textual information while preserving the time-evolving properties of entity embedding. (iii) To train the integration models, we construct three datasets, which align each quadruple with a relevant textual description, by adapting three existing temporal KG completion datasets. Extensive experiments show that ECOLA is model-agnostic and enhances temporal KG embedding models with up to **287%** relative improvements in the Hits@1 metric.

## 2 Preliminaries and Related Work

**Temporal Knowledge Graphs** Temporal knowledge graphs are multi-relational, directed graphs with labeled timestamped edges between entities (nodes). Let $\mathcal{E}$ and $\mathcal{P}$ represent a finite set of entities and predicates, respectively. tKG contains a collection of events written as quadruples. A quadruple $q = (e_s, p, e_o, t)$ represents a timestamped and labeled edge between a subject entity $e_s \in \mathcal{E}$ and an object entity $e_o \in \mathcal{E}$ at a timestamp $t \in \mathcal{T}$. Let $\mathcal{F}$ represents the set of all true quadruples, i.e., real events in the world, the temporal knowledge graph completion (tKGC) is the task of inferring $\mathcal{F}$ based on a set of observed facts $\mathcal{O}$, which is a subset of $\mathcal{F}$. Specifically, tKGC is to predict either a missing subject entity $(?, p, e_o, t)$ given the other three components or a missing object entity $(e_s, p, ?, t)$. We provide related works on temporal knowledge representations in Appendix A.

**Joint Language and Knowledge Models** Recent studies have achieved great success in jointly learning language and knowledge representations. Yamada et al. (2016) and Ganea & Hofmann (2017) use entity linking to map entities and words into the same representation space. Inspired by the success of contextualized language representation, Zhang et al. (2019) and Peters et al. (2019) focus on enhancing language models with external knowledge. They separately pre-train the entity embedding with knowledge embedding models, e.g., TransE Bordes et al. (2013), and inject the pre-trained entity embedding into PLMs, while fixing the entity embedding during training PLMs. Thus, they are not real joint models for learning the knowledge embedding and language embedding simultaneously. Yao et al. (2019), Kim et al. (2020), and Wang et al. (2021) learn to generate entity embeddings with pre-trained language models (PLMs) from entity descriptions. Moreover, He et al. (2019), Sun et al. (2020), and Liu et al. (2020) exploit the potential of contextualized knowledge representation by constructing subgraphs of structured knowledge and textual data instead of treating single triples as training units. Nevertheless, none of these works consider the temporal aspect of knowledge graphs, which makes them different from our proposed ECOLA.

## 3 ECOLA

In this section, we present the overall framework of ECOLA, including the model architecture in Section 3.1 - 3.3, a novel training task designed for aligning knowledge embedding and language representation in Section 3.4, and the training procedure in Section 3.5. As shown in Figure 2, ECOLA implicitly incorporates contextualized language representations into temporal knowledge embeddings by jointly optimizing the *knowledge-text prediction loss* and the *temporal knowledge embedding loss*. Note that, at inference time, we only take the enhanced temporal knowledge embeddings to perform the temporal KG completion task without using any textual data for preventing information leakage and keep fast inference speed.

### 3.1 Embedding Layer

In tKG embedding models, entity representations evolve over time. Thus, the key point of enhancing a time-dependent entity representation $\mathbf{e}_i(t)$ is to find texts that are relevant to the entity at the time of interest $t$. To this end, we use tKG quadruples (e.g., $(e_i, p, e_j, t)$) as an implicit measure for the alignment. We pair a quadruple with its relevant textual data and use such textual data to enhance

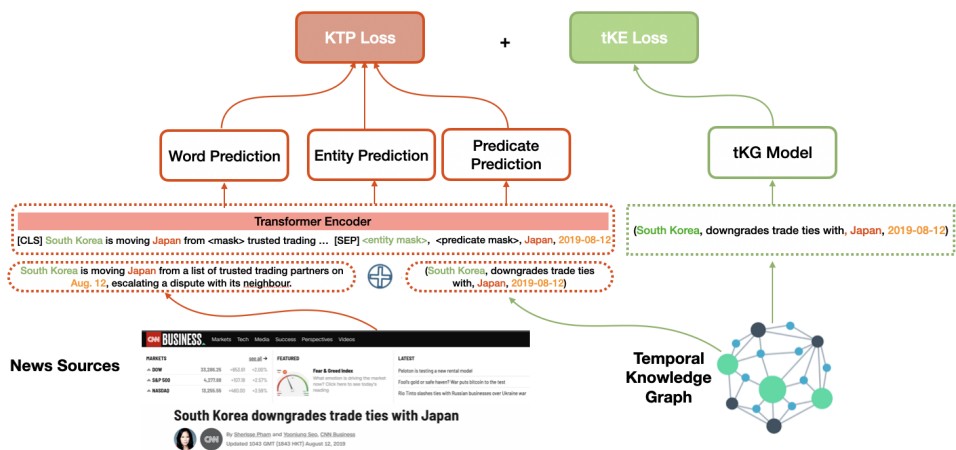

Figure 2: Model architecture. ECOLA jointly optimizes the knowledge-text prediction (KTP) objective and the temporal knowledge embedding (tKE) objective.

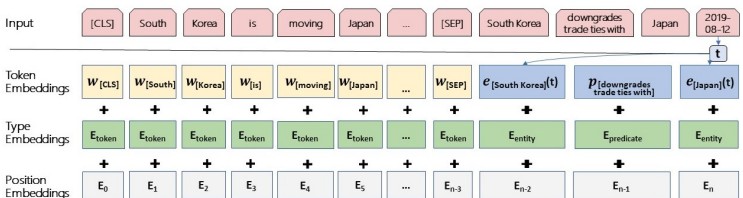

Figure 3: ECOLA input representations. Following textual tokens, the last four tokens in an input correspond to a quadruple from temporal knowledge graph. The last one is the timestamp $t$ for incorporating temporal information into entity representation $e_i(t)$. Here, $\mathbf{w}$ denotes subword token embedding, $\mathbf{e}$ and $\mathbf{p}$ denote entity and predicate embedding, respectively.

the entity representation $\mathbf{e}_i(t)$. Therefore, a training sample is a pair of a quadruple from temporal KGs and its corresponding textual description, which are packed together into a sequence. As shown in Figure 3, the input embeddings are the sum of token embedding, type embedding, and position embedding. For token embedding, we maintain **three lookup tables** for subwords, entities, and predicates, respectively. For subword embedding, we first tokenize the textual description into a sequence of subwords following Bert (Devlin et al., 2018) and use WordPiece embeddings (Wu et al., 2016) with a 30,000 token vocabulary. As the yellow tokens shown in the Figure 3, We denote a embedding sequence of subword tokens as $\{\mathbf{w}_1, ..., \mathbf{w}_n\}$. In contrast to subword embedding, the embeddings for entities and predicates are directly learned from scratch, similar to common knowledge embedding methods. We denote the entity embedding and predicate embedding as $\mathbf{e}$ and $\mathbf{p}$, respectively, as the blue tokens shown in Figure 3. We separate the knowledge tokens, i.e., entities and predicates, and subword tokens with a special token [SEP]. To handle different token types, we add type embedding to indicate the type of each token, i.e., subword, entity, and predicate. For position embedding, we assign each token an index according to its position in the input sequence and follow Devlin et al. (2018) to apply fully-learnable absolute position embeddings.

## 3.2 TEMPORAL KNOWLEDGE ENCODER

As shown in Figure 3, the input embedding for *entities* and *predicates* consists of knowledge token embedding, type embedding, and position embedding. In this section, we provide details of the temporal knowledge embedding objective.

A temporal embedding function defines entity embedding as a function that takes an entity and a timestamp as input and generates a time-dependent representation. There is a line of work exploring temporal embedding functions. Since we aim to propose a model-agnostic approach, we combine ECOLA with three temporal embedding functions, i.e., DyERNIE-Euclid (Han et al., 2020c), UTEE

(Han et al., 2021), and DE-SimplE (Goel et al., 2020). In the following, we refer to DyERNIE-Euclid as DyERNIE and take it as an example to introduce our framework. Specifically, the entity representation is derived from an initial embedding and a velocity vector

$$\mathbf{e}_i^{DyER}(t) = \bar{\mathbf{e}}_i^{DyER} + \mathbf{v}_{e_i} t,$$

where $\bar{\mathbf{e}}_i^{DyER}$ represents the initial embedding that does not change over time, and $\mathbf{v}_{e_i}$ is an entity-specific velocity vector. The combination with other temporal embedding functions are discussed in Section 4. The score function measuring the plausibility of a quadruple is defined as follows,

$$\phi^{DyER}(e_i, p, e_j, t) = -d(\mathbf{P} \odot \mathbf{e}_i^{DyER}(t), \mathbf{e}_j^{DyER}(t) + \mathbf{p}) + b_i + b_j, \tag{1}$$

where $\mathbf{P}$ and $\mathbf{p}$ represent the diagonal predicate matrix and the translation vector of predicate $p$, respectively, $d$ denotes the Euclidean distance, and $b_i$, $b_j$ are scalar biases of the subject and object $e_i$ and $e_j$ respectively. By learning tKE, we generate $M$ negative samples for each positive quadruple in a batch. We choose the binary cross entropy as the temporal knowledge embedding objective

$$\mathcal{L}_{tKE} = \frac{-1}{N} \sum_{k=1}^{N} (y_k \log(p_k) + (1 - y_k) \log(1 - p_k)), \tag{2}$$

where $N$ is the sum of positive and negative training samples, $y_k$ represents the binary label indicating whether a training sample is positive or not, $p_k$ denotes the predicted probability $\sigma(\phi_k^{DyER})$, and $\sigma(\cdot)$ represents the sigmoid function.

### 3.3 MASKED TRANSFORMER ENCODER

To encode the input sequence, we use the pre-trained contextual language representation model Bert (Devlin et al., 2018). Specifically, the encoder feeds a sequence of $N$ tokens including entities, predicates, and subwords into the embedding layer introduced in Section 3.1 to get the input embeddings and then computes $L$ layers of $d$-dimensional contextualized representations. Eventually, we get a contextualized representation for each token, which could be further used to predict masked tokens.

### 3.4 KNOWLEDGE-TEXT PREDICTION TASK

To incorporate textual knowledge into temporal knowledge embedding, we use the pre-trained language model Bert to encode the textual description and propose a knowledge-text prediction task to align the language representations and the knowledge embedding. The knowledge-text prediction task is an extension of the masked language modeling (MLM) task. As illustrated in Figure 2, given a pair of a quadruple and the corresponding event description, the knowledge-text prediction task is to randomly mask some of the input tokens and train the model to predict the original index of the masked tokens based on their contexts. As different types of tokens are masked, we encourage ECOLA to learn different capabilities:

- **Masking entities**. To predict an entity token in the quadruple, ECOLA has the following ways to gather information. First, the model can detect the textual mention of this entity token and determine the entity; second, if the other entity token and the predicate token are not masked, the model can utilize the available knowledge token to make a prediction, which is similar to the traditional semantic matching-based temporal KG models. Masking entity nodes helps ECOLA align the representation spaces of language and structured knowledge, and inject contextualized representations into entity embeddings.

- **Masking predicates**. To predict the predicate token in the quadruple, the model needs to detect mentions of subject entity and object entity and classify the semantic relationship between the two entity mentions. Thus, masking predicate tokens helps the model integrate language representation into the predicate embedding and map words and entities into a common representation space.

- **Masking subwords**. When subwords are masked, the objective is similar to traditional MLM. The difference is that ECOLA not only considers the dependency information in the text but also the entities and the logical relationship in the quadruple. Additionally, we initialize the encoder with the pre-trained BERT$_{\text{base}}$. Thus, masking subwords helps ECOLA keep linguistic knowledge and avoid catastrophic forgetting while integrating contextualized representations into temporal knowledge embeddings.

In each quadruple, the predicate and each entity have a probability of 15% to be masked. Similarly, we mask 15% of the subwords of the textual description at random. We ensure that entities and the predicate cannot be masked at the same time in a single training sample, where we conduct an ablation study in Section 6 to show the improvement of making this constraint. When a token is masked, we replace it with (1) the [MASK] token 80% of the time, (2) a randomly sampled token with the same type as the original token 10% of the time, (3) the unchanged token 10% of the time. For each masked token, the contextualized representation in the last layer of the encoder is used for three classification heads, which are responsible for predicting entities, predicates, and subword tokens, respectively. At last, a cross-entropy loss $\mathcal{L}_{KTP}$ is calculated over these masked tokens.

Although we focus on generating informative knowledge embeddings in this work, joint models often benefit both the language model and the temporal KG model. Unlike previous joint models (Zhang et al., 2019; Peters et al., 2019), we do not modify the Transformer encoder architecture, e.g., adding entity linkers or fusion layers. Thus, the language encoder enhanced by external knowledge can be adapted to a wide range of downstream tasks as easily as Bert. We evaluate the enhanced language model on the *temporal question answering* task and report the results in Appendix C.

### 3.5 TRAINING PROCEDURE AND INFERENCE

We initialize the transformer encoder with the pre-trained language model BERT$_{\text{base}}$[1] and the knowledge encoder with random vectors. Then we use the temporal knowledge embedding (tKE) objective $\mathcal{L}_{tKE}$ to train the knowledge encoder and use the knowledge-text prediction (KTP) objective $\mathcal{L}_{KTP}$ to incorporate temporal factual knowledge and textual knowledge in the form of a multi-task loss:

$$\mathcal{L} = \mathcal{L}_{tKE} + \lambda \mathcal{L}_{KTP},$$

where $\lambda$ is a hyperparameter to balance tKE loss and KTP loss. Note that those two tasks share the same embedding layer of entities and predicates. At inference time, we aim to answer link prediction queries, e.g., $(e_s, p, ?, t)$. Since there is no textual description at inference time, we take the entity and predicate embedding as input and use the score function of the knowledge encoder, e.g., Equation 1, to predict the missing links. Specifically, the score function assigns a plausibility score to each quadruple, and the proper object can be inferred by ranking the scores of all quadruples $\{(e_s, p, e_j, t), e_j \in \mathcal{E}\}$ that are accompanied with candidate entities.

## 4 MODEL VARIANTS

ECOLA is model-agnostic and can enhance different temporal knowledge embedding models. Besides ECOLA-DyERNIE, we introduce here two additional variants of ECOLA.

**ECOLA-DE** enhances the tKG embedding model DE-SimplE, which applies the diachronic embedding (DE) function (Goel et al., 2020). DE-function defines the temporal embeddings of entity $e_i$ at timestamp $t$ as

$$\mathbf{e}_i^{DE}(t)[n] = \begin{cases} \mathbf{a}_{e_i}[n] & \text{if } 1 \leqslant n \leqslant \gamma d, \\ \mathbf{a}_{e_i}[n] \sin(\boldsymbol{\omega}_{e_i}[n]t + \mathbf{b}_{e_i}[n]) \text{ else.} \end{cases} \tag{3}$$

Here, $\mathbf{e}_i^{DE}(t)[n]$ denotes the $n^{th}$ element of the embeddings of entity $e_i$ at time $t$. $\mathbf{a}_{e_i}, \boldsymbol{\omega}_{e_i}, \mathbf{b}_{e_i} \in \mathbb{R}^d$ are entity-specific vectors with learnable parameters, $d$ is the dimensionality of the embedding, and $\gamma \in [0, 1]$ represents the portions of the time-independent part. Besides, it use SimplE (Kazemi & Poole, 2018) as the score function of temporal knowledge embedding.

**ECOLA-UTEE** enhances UTEE Han et al. (2021) that learns a shared temporal encoding function for all entities to deal with the overfitting problem of the diachronic approach (Goel et al., 2020) on sparse datasets. Compared to ECOLA-DE, ECOLA-UTEE replaces Equation 3 with follows:

$$\mathbf{e}_i^{UTEE}(t) = [\bar{\mathbf{e}}_i || \mathbf{a} \sin(\boldsymbol{\omega} t + \mathbf{b})], \quad \bar{\mathbf{e}}_i \in \mathbb{R}^{\gamma d}; \mathbf{a}, \mathbf{w}, \mathbf{b} \in \mathbb{R}^{(1-\gamma)d}$$

where $\bar{\mathbf{e}}_i$ denotes entity-specific time-invariant part, the amplitude vector $\mathbf{a}$, frequency vector $\boldsymbol{\omega}$, and bias $\mathbf{b}$ are shared among all entities, $||$ denotes the concatenation operator, and $\gamma \in [0, 1]$.

---

[1]https://huggingface.co/bert-base-uncased

## 5 DATASETS

The training procedure of ECOLA requires both temporal knowledge graphs and textual descriptions. Given a quadruple $(e_s, p, e_o, t)$, the key point is to find texts that are temporally relevant to $e_s$ and $e_o$ at $t$. Existing temporal KG datasets do not provide such information. To facilitate the research on integrating textual knowledge into temporal knowledge embeddings, we reformat three existing datasets, i.e., GDELT[2], DuEE[3], and Wiki[4], for evaluating the proposed integration method. We show the statistics of the datasets in Table 2 in the appendix. Due to the limited size of upload files, we only attach DuEE and subsets of GDELT and Wiki in the supplementary material and will publish the complete version after acceptance.

**GDELT** is an initiative knowledge base storing events across the globe connecting people and organizations, e.g., (*Google, consult, the United States, 2018/01/06*). For each quadruple, GDELT provides the link to the news resource which the quadruple is extracted from. We assume each sentence that contains both mentions of subject entity and object entity is relevant to the given quadruple, and thus, temporally aligned with the subject and object at the given timestamp. We pair each of these sentences with the given quadruple to form a training sample. This process is similar to the distant supervision algorithm Mintz et al. (2009) in the relation extraction task, which assumes that, given a relationship between two entities, any sentence containing these two entities would express this relation. In total, the dataset contains 5849 entities, 237 predicates, 2403 timestamps, and 943956 quadruples with accompanying sentences.

**DuEE** is originally a human-annotated dataset for event extraction containing 65 event types and 121 argument roles. Each sample contains a sentence and several extracted event tuples. We reformat DuEE by manually converting event tuples into quadruples and then pairing the quadruples with their corresponding sentence.

**Wiki** is a temporal knowledge graph dataset proposed by Leblay & Chekol (2018), containing temporal facts from the Wikidata (Vrandečić & Krötzsch, 2014). Different from GDELT and DuEE, time annotations in Wiki are represented as time intervals, e.g., (Savonranta, instance of, municipality of Finland, 1882 - 2009). Following the post-processing by Dasgupta et al. (2018), we discretize the time span into 82 different timestamps. We align each entity to its Wikipedia page and extract the first abstract section as its description. To construct the relevant textual data of each quadruple, we combine the subject entity description, relation, and object description into a sequence, separated by [SEP] token between two neighboring parts. In this case, the knowledge-text prediction task let subject entity see the descriptions of its *neighbors at different timestamps*, and thus, preserving the temporal alignment between time-dependent entity representation and textual knowledge.

## 6 EXPERIMENTS

We evaluate the enhanced temporal knowledge embedding on the temporal KG completion task. Specifically, we take the entity and predicate embedding of ECOLA-DyERNIE and use Equation 1 to predict missing links. To make a fair comparison with other temporal KG embedding models, we do not use any textual descriptions at test time.

**Baselines** We include both static and temporal KG embedding models. From the static KG embedding models, we use TransE (Bordes et al., 2013), DistMult (Yang et al., 2014), and SimplE (Kazemi & Poole, 2018). These methods ignore the time information. From the temporal KG embedding models, we compare our model with several state-of-the-art methods, including AiTSEE (Xu et al., 2019), TNTComplE(Lacroix et al., 2020), DyERNIE[5] (Han et al., 2020c), TeRO (Xu et al., 2020), and DE-SimplE (Goel et al., 2020). We provide implementation details in Appendix D and attach the source code in the supplementary material.

---

[2]https://www.gdeltproject.org/data.html#googlebigquery

[3]https://ai.baidu.com/broad/download

[4]https://www.wikidata.org/wiki/Wikidata:Main_Page

[5]For a fair comparison with other baselines, we choose DyERNIE-Euclid.

**Evaluation protocol** For each quadruple $q = (e_s, p, e_o, t)$ in the test set $\mathcal{G}_{test}$, we create two queries: $(e_s, p, ?, t)$ and $(?, p, e_o, t)$. For each query, the model ranks all possible entities $\mathcal{E}$ according to their scores. Following the filtered setting in Han et al. (2020b), we remove all entity candidates that correspond to true triples from the candidate list apart from the current test entity. Let $Rank(e_s)$ and $Rank(e_o)$ represent the rank for $e_s$ and $e_o$ of the two queries respectively, we evaluate our models using standard metrics across the link prediction literature: *mean reciprocal rank (MRR)*: $\frac{1}{2 \cdot |\mathcal{G}_{test}|} \sum_{q \in \mathcal{G}_{test}} \left( \frac{1}{Rank(e_s)} + \frac{1}{Rank(e_o)} \right)$ and *Hits@k* ($k \in \{1, 3, 10\}$): the percentage of times that the true entity candidate appears in the top $k$ of ranked candidates.

Table 1: Temporal link prediction results: Mean Reciprocal Rank (MRR, %) and Hits@1/3(%). The results of the proposed fusion models (in bold) and their counterpart KG models are listed together.

| Datasets | GDELT - filtered | | | Wiki - filtered | | | DuEE - filtered | | |
|---|---|---|---|---|---|---|---|---|---|
| Model | MRR | Hits@1 | Hits@3 | MRR | Hits@1 | Hits@3 | MRR | Hits@1 | Hits@3 |
| TransE | 8.08 | 0.00 | 8.33 | 27.25 | 16.09 | 33.06 | 34.25 | 4.45 | 60.73 |
| SimplE | 10.98 | 4.76 | 10.49 | 20.75 | 16.77 | 23.23 | 51.13 | 40.69 | 58.30 |
| DistMult | 11.27 | 4.86 | 10.87 | 21.40 | 17.54 | 23.86 | 48.58 | 38.26 | 55.26 |
| TeRO | 6.59 | 1.75 | 5.86 | 32.92 | 21.74 | 39.12 | 54.29 | 39.27 | 63.16 |
| ATiSE | 7.00 | 2.48 | 6.26 | 35.36 | 24.07 | 41.69 | 53.79 | 42.31 | 59.92 |
| TNTComplEx | 8.93 | 3.60 | 8.52 | 34.36 | 22.38 | 40.64 | 57.56 | 43.52 | 65.99 |
| UTEE | 9.76 | 4.23 | 9.77 | 26.96 | 20.98 | 30.39 | 53.36 | 43.92 | 60.52 |
| **ECOLA-UTEE** | 19.11 $\pm$ 00.16 | 15.29 $\pm$ 00.38 | 19.46 $\pm$ 00.05 | 38.35 $\pm$ 00.22 | 30.56 $\pm$ 00.18 | 42.11 $\pm$ 00.14 | **60.36** $\pm$ 00.36 | **46.55** $\pm$ 00.51 | 69.22 $\pm$ 00.93 |
| DyERNIE | 10.72 | 4.24 | 10.81 | 23.51 | 14.53 | 25.21 | 57.58 | 41.49 | **70.24** |
| **ECOLA-DyERNIE** | **19.99** $\pm$ 00.05 | **16.40** $\pm$ 00.09 | **19.78** $\pm$ 00.03 | **41.22** $\pm$ 00.04 | **33.02** $\pm$ 00.06 | **45.00** $\pm$ 00.27 | 59.64 $\pm$ 00.20 | 46.35 $\pm$ 00.18 | 67.87 $\pm$ 00.53 |

**Quantitative Study** Table 1 reports the tKG completion results on the test sets, which are averaged over three trials. The error bars of the ECOLA models are also provided. Firstly, we can see that ECOLA-UTEE improves its baseline temporal KG embedding model, UTEE, by a large margin, demonstrating the effectiveness of our fusing strategy. Specifically, ECOLA-UTEE enhances UTEE on GDELT with a *relative improvement* of **95%** and **99%** in terms of mean reciprocal rank (MRR) and Hits@3, even nearly **four times** better in terms of Hits@1. Thus, its superiority is clear on GDELT, which is the most challenging dataset among benchmark tKG datasets, containing nearly one million quadruples and more than two hundred relations. Secondly, ECOLA-UTEE and ECOLA-DE generally outperform UTEE and DE-SimplE on the three datasets, demonstrating that ECOLA is model-agnostic and able to enhance different tKG embedding models. Besides, in the DuEE dataset, ECOLA-DyERNIE achieves a better performance than DyERNIE in Hits@1 and MRR, but the gap reverses in Hits@3. The reason could be that ECOLA-DyERNIE is good at classifying hard negatives using textual knowledge, and thus has a high Hits@1; however, since DuEE is much smaller than the other two datasets, ECOLA-DyERNIE may overfit in some cases, where the ground truth is pushed away from the top 3 rank.

We compare the performance of DE-SimplE, ECOLA-DE, and ECOLA-SF on GDELT in Figure 4a. ECOLA-SF is the **static counterpart** of ECOLA-DE, where we do not consider the temporal alignment while incorporating textual knowledge. Specifically, ECOLA-SF integrates all textual knowledge into the **time-invariant part** of entity representations. We provide more details of ECOLA-SF in Appendix B. We can see ECOLA-DE significantly outperforms DE-SimplE and ECOLA-SF in terms of MRR and Hits@1. In particular, the performance gap between ECOLA-DE and ECOLA-SF is significant, demonstrating the *temporal alignment* between time-dependent entity representation and textual knowledge is more powerful than the *static alignment*.

**Ablation study** Figure 4b shows the results of different masking strategies on GDELT. The first strategy called *Masking E+R+W* which allows to simultaneously mask predicate, entity, and subword tokens in the same training sample. The second strategy is *Masking E/R+W*, where we mask 15% subword tokens in the language part, and either an entity or a predicate in the knowledge tuple. In other words, simultaneously masking an entity and a predicate in a training sample is not allowed. In the third strategy called *Masking E/R/W*, for each training sample, we choose to mask either subword

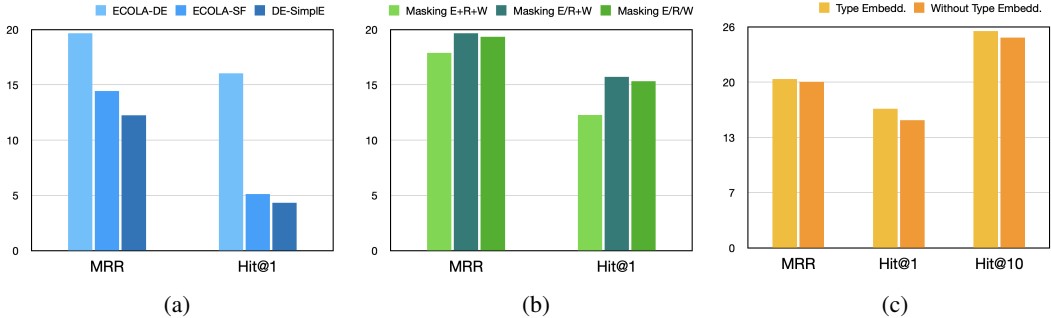

Figure 4: Ablation Study. (a) Temporal alignment analysis. We compare De-SimplE, ECOLA-DE, and ECOLA-SF in terms of MRR(%) and Hits@1(%) on GDELT. (b) Masking strategy analysis. We compare ECOLA-DE with different masking strategies and show the results of MRR(%) and Hits@1(%) on GDELT. (c) Type embedding analysis. We compare ECOLA-DE with/without type embedding and show the results of MRR(%) and Hits@1/10(%) on GDELT.

tokens, an entity, or the predicate. The experimental results show the advantage of the second masking strategy, indicating that remaining adequate information in the knowledge tuple helps the model to align the knowledge embedding and language representations. Figure 4c demonstrates the ablation study on the type embedding, which differentiates among subword tokens, entity, and predicate of the input. We can observe that removing type embedding leads to a considerable performance gap on GDELT, indicating that providing distinguishment between subwords, entities, and predicates helps the model to better understand different input components and different prediction tasks.

**Qualitative Analysis**    To investigate why incorporating textual knowledge can improve the tKG embedding models' performance, we study the test samples that have been correctly predicted by the fusion model ECOLA-DE but wrongly by the tKG model DE-SimplE. It is observed that language representations help overcome the incompleteness of the tKG by leveraging knowledge from augmented textual data. For example, there is a test quadruple *(US, host a visit, ?, 19-11-14)* with ground truth *R.T. Erdoğan*. The training set contains a quite relevant quadruple, i.e., *(Turkey, intend to negotiate with, US, 19-11-11)*. However, the given tKG does not contain information indicating that the entity *R.T. Erdoğan* is a representative of *Turkey*. So it is difficult for the tKG model DE-SimplE to infer the correct answer from the above-mentioned quadruple. In ECOLA-DE, the augmented textual data do contain such information, e.g. *"The president of Turkey, R.T. Erdogan, inaugurated in Aug. 2014."*, which narrows the gap between *R.T. Erdogan* and *Turkey*. Thus, by integrating textual information into temporal knowledge embedding, the enhanced model can gain additional information which the knowledge base does not include. Another example is relevant to the entity *Charles de Gaulle*. To infer the test quadruple *(Charles de Gaulle, citizenship of, ?, 1958)* with ground truth *French 5$^{th}$ Republic*, it is noticed that in the training set of ECOLA-DE, we have quadruple *(Charles de Gaulle, president of, French 4$^{th}$ Republic, 1957)* with supporting textual data *"Charles de Gaulle was the last president of French 4$^{th}$ Republic, and French 5$^{th}$ Republic emerged from the collapse of the 4$^{th}$ Republic in 1958."*, which shows that the entity representation of *Charles de Gaulle* is enhanced by the evolving history of France and is temporally closer to *French 5$^{th}$ Republic* at the query timestamp *1958*.

## 7    CONCLUSION

We propose ECOLA to enhance temporal knowledge embedding using textual knowledge. Specifically, we enhance time-evolving entity representations with temporally relevant textual data by encoding the textual data using a pre-trained language model and introducing a novel knowledge-text prediction task to align the temporal knowledge and language representation into the same semantic space. Besides, we construct three datasets that contain paired structured temporal knowledge and unstructured textual descriptions, which can benefit future research on fusing temporal structured and unstructured knowledge. Extensive experiments show ECOLA is model-agnostic and can improve many temporal knowledge graph models by a large margin.

**Reproducibility Statement** About **datasets**, Since the size of GDELT and Wiki exceeds the limit of upload file size allowed by ICLR, we only upload DuEE, partial Wiki and partial GDELT (short version with 1000 samples) in the supplementary material due to the and will publish the complete dataset of Wiki and GDELT after acceptance. Besides, we provide the description of the data processing steps in Section 5. We provide the dataset statistics in Table 2 in appendix. Additionally, we provide our source code in the supplementary material.

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

## APPENDIX

Table 2: Datasets Statistics

| Dataset | # Entities | # Predicates | # Timestamps | # training set | # validation set | # test set |
|---------|-----------|--------------|--------------|----------------|------------------|------------|
| GDELT   | 5849      | 237          | 2403         | 755166         | 94395            | 94395      |
| DUEE    | 219       | 41           | 629          | 1879           | 247              | 247        |
| WIKI    | 10844     | 23           | 82           | 233525         | 19374            | 19374      |

Table 3: Hyperparameter settings of ECOLA and baselines.

| Parameters | Embedding dimension | | | Negative Sampling | | | Learning rate | | | Batch Size | | |
|------------|------|------|------|------|------|------|------|------|------|------|------|------|
| Datasets | GDELT | DuEE | Wiki | GDELT | DuEE | Wiki | GDELT | DuEE | Wiki | GDELT | DuEE | Wiki |
| TransE | 768 | 768 | 768 | 200 | 100 | 100 | 5e-4 | 5e-4 | 5e-4 | 256 | 128 | 256 |
| SimplE | 768 | 768 | 768 | 200 | 100 | 100 | 5e-4 | 5e-4 | 5e-4 | 256 | 128 | 256 |
| TTransE | 768 | 768 | 768 | 200 | 100 | 100 | 5.2e-4 | 5.2e-4 | 5.2e-4 | 256 | 256 | 256 |
| TNTComplEx | 768 | 768 | 768 | 200 | 100 | 100 | 1.5e-4 | 1.5e-4 | 1.5e-4 | 256 | 256 | 256 |
| DE-SimplE | 768 | 768 | 768 | 200 | 100 | 100 | 5e-4 | 5e-4 | 5e-4 | 256 | 128 | 256 |
| ECOLA-SF | 768 | 768 | 768 | 200 | 100 | 100 | 1e-4 | 2e-5 | 1e-4 | 64 | 16 | 64 |
| ECOLA-DE | 768 | 768 | 768 | 200 | 200 | 200 | 2e-5 | 2e-5 | 2e-5 | 4 | 8 | 4 |
| ECOLA-UTEE | 768 | 768 | 768 | 200 | 200 | 200 | 2e-5 | 2e-5 | 2e-5 | 4 | 8 | 4 |
| ECOLA-dyERNIE | 768 | 768 | 768 | 200 | 200 | 200 | 2e-5 | e-4 | 2e-5 | 4 | 8 | 4 |

## A RELATED WORK OF TEMPORAL KNOWLEDGE EMBEDDING

Temporal Knowledge Embedding (tKE) is also termed as Temporal Knowledge Representation Learning (TKRL), which is to embed entities and predicates of temporal knowledge graphs into low-dimensional vector spaces. TKRL is an expressive and popular paradigm underlying many KG models. To capture temporal aspects, each model either embeds discrete timestamps into a vector space or learns time-dependent representations for each entity. Ma et al. (2019) developed extensions of static knowledge graph models by adding timestamp embeddings to their score functions. Besides, HyTE (Dasgupta et al., 2018) embeds time information in the entity-relation space by learning a temporal hyperplane to each timestamp and projects the embeddings of entities and relations onto timestamp-specific hyperplanes. Later, Goel et al. (2020) equipped static models with a diachronic entity embedding function which provides the characteristics of entities at any point in time and achieves strong results. Moreover, Han et al. (2020c) introduced a non-Euclidean embedding approach that learns evolving entity representations in a product of Riemannian manifolds. It is the first work to contribute to geometric embedding for tKG and achieves state-of-the-art performances on the benchmark datasets. In particular, ECOLA is model-agnostic, which means any temporal KG embedding model can be potentially enhanced by training with the knowledge-text task.

## B ECOLA-SF: AN ABLATION STUDY ON STATIC FUSION

We compare the effectiveness of enhancing temporal knowledge embedding and enhancing static knowledge embedding. In particular, we only feed the static part of entity embeddings into PLM to perform the knowledge-text prediction task. We refer it as ECOLA-SF (**S**tatic**F**usion).

**ECOLA-SF** is the static counterpart of ECOLA-DE, where we do not apply temporal knowledge embedding to the knowledge-text prediction objective $\mathcal{L}_{KTP}$. Specifically, we randomly initialize an embedding vector $\bar{\mathbf{e}}_i \in \mathbb{R}^d$ for each entity $e_i \in \mathcal{E}$, where $\bar{\mathbf{e}}_i$ has the same dimension as the token embedding in pre-trained language models. Then we learn the **time-invariant part** $\bar{\mathbf{e}}_i$ via the knowledge-text prediction task. For the tKE objective, we have the following temporal knowledge embedding,

$$\mathbf{e}_i^{SF}(t)[n] = \begin{cases} \mathbf{W}_{sf}\bar{\mathbf{e}}_i[n] & \text{if } 1 \leqslant n \leqslant \gamma d, \\ \mathbf{a}_{e_i}[n]\sin(\boldsymbol{\omega}_{e_i}[n]t + \mathbf{b}_{e_i}[n]) & \text{else}, \end{cases}$$

where $\mathbf{e}_i^{SF}(t) \in \mathbb{R}^d$ is an entity embedding containing static and temporal embedding parts. $\mathbf{a}_{e_i}, \boldsymbol{\omega}_{e_i}, \mathbf{b}_{e_i} \in \mathbb{R}^{d-\gamma d}$ are entity-specific vectors with learnable parameters. $\mathbf{W}_{sf} \in \mathbb{R}^{d \times \gamma d}$ is

| Model | Hits@1 (%) | Hits@10 (%) |
|---|---|---|
| CRONKGQA | 25.8 | 52.0 |
| ECOLA-CRONKGQA | 27.5 | 54.4 |

Table 4: Performance of ECOLA enhaned language model in tKBQA task.

matrix with learnable weights. Note that $\mathbf{e}_i^{SF}(t)$ only plays a role in $\mathcal{L}_{tKE}$, and we use static embedding $\bar{\mathbf{e}}_i$ instead of $\mathbf{e}_i^{SF}(t)$ in $\mathcal{L}_{KTP}$.

## C ENHANCEMENT ON LANGUAGE REPRESENTATIONS

Although our work focuses on enhancing temporal knowledge embeddings with contextualized language representations, joint models often benefit both the language model and the tKG model due to the mutual information exchange between language and tKGs during joint training. To study ECOLA's enhancement on the language model, we selected temporal question answering as a downstream task to show that the proposed ECOLA can also benefit the language model.

**Temporal Question Answering over Temporal Knowledge Graphs (TKGQA)** Natural questions often include temporal constraints, e.g., who was the US president before Jimmy Carter? To deal with such challenging temporal constraints, temporal question answering over temporal knowledge base, formulated as TKGQA task, has become trendy since tKGs help to find entity or timestamp answers with support of temporal facts.

**Performance Gain on TKGQA** Saxena et al. (2021a) introduced the TKGQA dataset CRONQUES-TIONS containing natural temporal questions with different types of temporal constraints and an accompanying temporal knowledge graph (tKG). They proposed a baseline called CRONKGQA that uses a pre-trained language model (BERT) to understand the implicit representation of temporal constraints in temporal questions followed by a scoring function for answer prediction. We enhance the language encoder in CRONKGQA with the proposed ECOLA approach, i.e., we find temporal relevant texts for quadruples in the supporting tKG given in CRONQUESTIONS and train the language model and the tKG model jointly with the proposed KTP task. Then we plug the enhanced language model back into CRONKGQA and name the enhanced model as ECOLA-CRONKGQA. The models are evaluated with standard metrics $Hits@k(k \in \{1,3\})$: the percentage of times that the true entity or time candidate appears in the top $k$ of ranked candidates. As shown in Table 4, empirical results show that our proposed ECOLA enhances the language model with **7.4 % relative improvements** regarding precision on CRONQUESTIONS, demonstrating the benefits of ECOLA to the language model.

## D IMPLEMENTATION

We use the datasets augmented with reciprocal relations to train all baseline models. We tune hyperparameters of our models using the random search and report the best configuration. Specifically, we set the loss weight $\lambda$ to be 0.3, except for ECOLA-DE model trained on Wiki dataset where $\lambda$ is set to be 0.001. We use the Adam optimizer (Kingma & Ba, 2014). We use the implementation of DE-SimplE[6], ATiSE/TeRO[7]. We use the code for TNTCopmlEx from the tKG framework (Han et al., 2021). We implement TTransE based on the implementation of TransE in PyKEEN[8]. We provide the detailed settings of hyperparameters of each baseline model and ECOLA in Table 3 in the appendix.

## E THE AMOUNT OF COMPUTE AND THE TYPE OF RESOURCES USED

We run our experiments on an NVIDIA A40 with a memory size of 48G. We provide the **training time** of our models and some baselines in Table 5. Note that there are no textual descriptions at

---

[6]https://github.com/BorealisAI/de-simple
[7]https://github.com/soledad921/ATISE
[8]https://github.com/pykeen/pykeen

inference time, and we take the entity and predicate embedding as input and use the score function of KG models to predict the missing links. Thus, the inference time of ECOLA (e.g., ECOLA-DE) and its counterpart KG model (e.g., DE-SimplE) is the **same**.

Table 5: The runtime of training procedure (in hours).

| Dataset | GDELT | DuEE | Wiki |
|---|---|---|---|
| DE-SimplE | 17 | 0.5 | 5.0 |
| ECOLA-DE | 24.0 | 16.7 | 43.2 |
| UTEE | 67.3 | 0.5 | 11.3 |
| ECOLA-UTEE | 36.0 | 12.8 | 45.6 |
| DyERNIE | 25 | 0.1 | 5.9 |
| ECOLA-DyERNIE | 23.8 | 10.8 | 67.2 |

