# OpenReview forum: "Enhanced Temporal Knowledge Embeddings with Contextualized Language Representations"
_ICLR.cc/2023/Conference — Submitted to ICLR 2023_

### Official Review · Reviewer_r4WS · 2022-10-16

**Confidence:** 3
**Correctness:** 3
**Technical Novelty And Significance:** 3
**Empirical Novelty And Significance:** 3
**Recommendation:** 6

**Clarity, Quality, Novelty And Reproducibility:**

- Clarity - The writing and presentation are clear
- Quality - The paper is well-executed
- Novelty - I think the novelty is sufficient. The method is a relatively intuitive extension of text-KG models to the temporal setting, but there is a sufficient contribution in designing a method such that it actually improves performance on temporal KG tasks.
- Reproducibility - the authors promise to release data and code, which is great.

**Strength And Weaknesses:**

Strength
- The paper is generally well-written and easy to understand
- The problem studied is well-motivated. Static KGs are often studied in existing works, but a lot of knowledge is inherently time-dependent, motivating temporal KG representations. The use of textual information for temporal KG is an important direction for coping with the sparsity of tKGs.
- The paper identifies clear technical challenges (alignment of text and KG tuples, and modeling of temporal information), and proposes reasonable solutions to them.
- Experiments are solid, conducted on three datasets, and show that the proposed method provides significant improvement in performance.

Weakness / suggestion for improvement
- The authors mention that at inference time, the method does not use textual descriptions for test quadruples, but it still achieves improved performance. I wonder if the authors could provide a bit more discussions about the intuition behind this? - e.g. Is this because the test entities were seen during training and the text information used during training already provided extra knowledge for these test entities? A related question is, would the proposed method work if an unseen entity is given at test time? I think it would be great if the authors could add these discussions to the paper.

Question
- Did the author also consider masking and predicting the temporal information in the KTP task? If not, might that be an interesting thing to try?

Typo
- Page 2 at “a novel knoweldge-text prediction (KTP) task”: “knoweldge” -> “knowledge”

**Summary Of The Paper:**

- This paper studies temporal knowledge graph (tKG) embeddings and proposes a new method that leverages textual knowledge/language models to overcome the sparsity of KGs. The authors identify two challenges: aligning temporally relevant textual information with KG entities, and incorporating temporal information into joint text-KG modeling. The paper then proposes the ECOLA method that addresses these two challenges: it creates new datasets that align tKG and text using using tKG quadruples, and uses a knowledge-text prediction (KTP) task to combine textual knowledge and temporal knowledge quadruples from tKG. The proposed method outperforms existing tKG embedding methods that do not use textual information.

**Summary Of The Review:**

Overall, the paper studies an important problem and develops a method that works well. There is some room for improvements, but I did not find critical weaknesses. Overall I think this paper is above acceptance threshold.

---

> ### Author Response · Authors · 2022-11-18
> **Response to Reviewer r4WS, PartA**
>
> **Thank you very much for your effort of reviewing our paper and providing valuable comments and suggestions. We appreciate it. We try to solve your concerns in the following.**
>
> **Weakness / suggestion for improvement:** *the authors mention that at inference time, the method does not use textual descriptions for test quadruples, but it still achieves improved performance. I wonder if the authors could provide a bit more discussions about the intuition behind this? - e.g. Is this because the test entities were seen during training and the text information used during training already provided extra knowledge for these test entities? A related question is, would the proposed method work if an unseen entity is given at test time? I think it would be great if the authors could add these discussions to the paper.*
>
> **Reply:** Yes, we conduct transductive learning on temporal knowledge graphs and all test entities were seen during training. Since we do not use textual descriptions at test time, the main reason for the performance enhancement would be that the test entities were seen during training and got enhanced by relevant textual information during the training. As stated in the paragraph titled *Qualitative Analysis* in Section 6 in our manuscript, the textual information used during training helps overcome the incompleteness of the tKG and thus improve the quality of knowledge embedding, i.e., entities and predicates. For example, there is a **test quadruple** *(US, host a visit, ?, 19-11-14)* with ground truth *R.T. Erdogan*. The **training set** contains a quite relevant quadruple, i.e., *(Turkey, intend to negotiate with, US, 19-11-11)*. However, the given tKG does not contain information indicating that the entity *R.T. Erdogan* is a representative of *Turkey*. So it is difficult for the tKG model DE-SimplE to infer the correct answer based on the above-mentioned training quadruple. In our proposed ECOLA, the augmented textual data do contain such information, e.g. "*The president of Turkey, R.T. Erdogan, inaugurated in Aug. 2014.*", which narrows the gap between *R.T. Erdogan* and *Turkey*, e.g., reduces the distance between these two entities, such that the model can predict that US host a visit of Erdogan in Nov. 2019. Thus, by integrating extra textual information at training, the temporal knowledge embedding can gain additional knowledge that the knowledge base does not include and become more expressive.
>
> Regarding your question about whether our proposed method work if an unseen entity is given at test time, our answer would be yes. If an unseen entity is given at test time, our proposed method can still benefit the inference by combining with inductive representation learning approaches, e.g., [1]. The intuition would be that we could estimate the representations of an unseen test entity using its temporal neighbors which have been seen at training. Thus, we could implicitly improve the representation of the unseen test entity by learning a better representation of its temporal neighbors.
>
> [1] Zifeng Ding, Jingpei Wu, Bailan He, Yunpu Ma, Zhen Han, Volker Tresp. Few-Shot Inductive Learning on Temporal Knowledge Graphs using Concept-Aware Information. In Proceeding of the 2022 Conference on Automated Knowledge Base Construction (AKBC), 2022.

---

> > ### Author Response · Authors · 2022-11-18
> > **Response to Reviewer r4WS, Part B**
> >
> > **Question**: *Did the author also consider masking and predicting the temporal information in the KTP task? If not, might that be an interesting thing to try?*
> >
> > **Reply**: Thank you for making an excellent point about strengthening our model.  And **yes**, we did such experiments.
> >
> > Driven by the motivation of studying the effect of masking temporal information as purely as possible, we choose to mask temporal information in ECOLA-UTEE since the time embedding in UTEE is not entity-specific but shared among all entities. Simply put, we extend the existing KTP task which consists of three subtasks to four subtasks, i.e, word prediction, entity prediction, relation prediction, and one more time prediction task, where the timestamp in the input samples are masked randomly, and the model learn to predict the correct original timestamp.
> >
> > Correspondingly, the input for the extended KTP task will be appended with a timestamp token, such as “[CLS] ….word tokens … [SEP] [subject] [predicate] [object] [timestamp]”.  The designed input embedding remains to be the addition of token embedding, type embedding, and position embedding, except for that: firstly, the token embedding sequence is appended with time embedding from UTEE model**;** secondly, the type embedding is extended from three types to four types, with one more type representing time embedding. Then, the model jointly learns the language representations and temporal knowledge embeddings through the extended KTP task addressing time information compared to the previous version.
> >
> > The results of masking time information in ECOLA-UTEE data have significant performance gain regarding metrics MRR and Hits@K with MRR = 20.39, Hits@1 = 16.83, Hits@3 = 20.08, which is **6.7% relatively higher** on MRR, 1**0.1% relatively higher** on Hits@1 and **3.2% relatively higher** on Hits@3 respectively. We conjecture that the additional time prediction task enhances the time representation due to the disentanglement from time-dependent entity representation learning. For now, we leave this exciting finding to future work for further inspections.
> >
> > However, it’s difficult to develop a model-agnostic approach to mask and predict the temporal information since different temporal knowledge embedding methods model the temporal information in different ways. For example, DE-SimplE [2] covers the temporal information directly into the entity representation vector $\mathbf e \in \mathbb R^d$, where the first $\gamma d$  elements of the vector are time-evolving, and the other $(1 − \gamma)d$ elements are time-invariant with $0 \le \gamma \le 1$.  In DyERNIE [3], the entity representation $\mathbf e \in \mathbb R^d$ is derived from an initial embedding $\bar{\mathbf e} in \mathbb R^d$ and an entity-specific velocity vector $\mathbf v_e \in  \mathbb R^d$ to encode both the stationary properties and the temporal information. HyTE [4] explicitly incorporates the temporal information in the entity-relation space by associating each timestamp with a corresponding hyperplane. The entity representation at a specific timestamp is the projection of the initial entity representation onto the corresponding timestamp-specific hyperplane. Thus, masking and predicting temporal information will be specific to each temporal knowledge embedding model.
> >
> > [2] Rishab Goel, Seyed Mehran Kazemi, Marcus Brubaker, and Pascal Poupart. Diachronic embedding for temporal knowledge graph completion. In Proceedings of the AAAI Conference on Artificial Intelligence, 2020.
> > [3] Zhen Han, Peng Chen, Yunpu Ma, Volker Tresp. DyERNIE: Dynamic Evolution of Riemannian Manifold Embeddings for Temporal Knowledge Graph Completion. In Proceedings of the Conference on Empirical Methods in Natural Language Processing (EMNLP), 2020.
> > [4] Shib Sankar Dasgupta, Swayambhu Nath Ray, and Partha Talukdar. Hyte: Hyperplane-based temporally aware knowledge graph embedding. In Proceedings of the 2018 conference on empirical methods in natural language processing, pp. 2001–2011, 2018.
> >
> > **Typo**
> >
> > *- Page 2 at “a novel knoweldge-text prediction (KTP) task”: “knoweldge” -> “knowledge”*
> >
> > **Reply:** Thank you for pointing it out. We have corrected it in the updated version.

---

### Official Review · Reviewer_cTPN · 2022-10-25

**Confidence:** 4
**Clarity, Quality, Novelty And Reproducibility:** 1. The paper is clear and well-writte…
**Correctness:** 3
**Technical Novelty And Significance:** 2
**Empirical Novelty And Significance:** 3
**Recommendation:** 6

**Strength And Weaknesses:**

Strength:

1. The paper is well-written and easy to follow. The motivation to consider time-evolving relations in KGs is clear. Leveraging the rich textual information to enhance the embeddings of knowledge graph is interesting.

2. Propose a simple yet effective method to improve the representations of tKGs and the experiment results show the ECOLA can improve embeddings from various embedding methods and achieve large improvement.

Weakness:

1. Novelty is the main concern. The main method ECOLA extends the MLM objective to predict both masked tokens and entities or relationships. The Model Variants are merely combinations of ECOLA with different previous knowledge embedding methods.

2. Not clear what benefits the language model gain after the joint training. The paper can be stronger if the author can show that the proposed ECOLA can benefit the language model as well.

**Summary Of The Paper:**

This paper mainly addresses the problem of learning the representations of temporal knowledge graphs (tKGs) which contain time-evolving relations between the subject and object. To alleviate the sparsity of the tKG and learn better embeddings, the authors propose to extend the MLM objective so that during training, the knowledge quadruples are paired with corresponding texts and some of the tokens or entities are masked. The model is trained to predict the mask tokens and the mask entities or relationships. This way the model can build latent connections between two embeddings spaces.

**Summary Of The Review:**

Overall the paper is clear and well-motivated. It has strong empirical results and the results can support the claim that the proposed method is model-agnostic. Some analysis is conducted and provides some insights. Some of the remaining research questions need to be answered (see questions below).

Question:
1. What do you mean by “we solve the temporal alignment challenge by using tKG quadruples as an implicit measure. ” Can you be more specific with the term “implicit measure”?

2. Based on the results we can see that the ECOLA mainly leverages the language model to improve the KG embeddings. I wonder if the performance can be further improved if larger language models are leveraged since BERT base is relatively small.

3. Does the quality of the text significantly affect the model’s performance? If in another domain there is no high-quality paired text, can the proposed method still improve the KG embeddings?

---

> ### Author Response · Authors · 2022-11-14
> **Response to Reviewer cTPN, part A**
>
> **We want to say thank you for taking the time to review our paper and provide valuable feedback. We would like to address your concerns and questions in the following.**
>
> **Question 1**:  What do you mean by “we solve the temporal alignment challenge by using tKG quadruples as an implicit measure. ” Can you be more specific with the term “implicit measure”?
>
> **Reply**: Before explaining the term “implicit measure,” we would like first to point out the challenge when using textual knowledge to enhance temporal knowledge embedding. Existing approaches such as KG-Bert [1] and KEPLER [2] often use a time-invariant entity description to boost the entity embedding in KGs. However, in temporal knowledge graphs, entities usually have an inherent temporal nature, i.e., their characteristics evolve with time, and they may form different relationships with other entities at different timestamps. Thus, given an entity, it should be taken into account which textual knowledge is relevant to it at which timestamp. We name this challenge temporal alignment between **textual knowledge** and **entity embedding**. A straightforward way to solve this challenge is to find temporally-valid entity descriptions, which describe the status of an **entity** at a specific timestamp. It can be seen as a naive extension of KG-Bert and KEPLER. However, it is difficult in practice to directly obtain the required temporally-valid entity descriptions for an entity. Therefore, we turn to an implicit way inspired by our findings: we find that many temporal knowledge graphs such as GDELT [3] provide the news resource where a quadruple is extracted. In other words, we know which texts are relevant to a quadruple, i.e., (subject $s$, predict $p$, object $o$, timestamp $t$). If a text is relevant to a quadruple, then it should also be **temporally relevant** to the $s$ and $o$ at $t$. We then choose to use event descriptions to enhance entity embedding. In a word, we first find texts that are relevant to a quadruple and then use such texts to enhance entities involved in the quadruple. Therefore, quadruples play a role as a bridge that helps to find texts that are temporally relevant to an entity at a specific timestamp. This is what we mean “implicit measure” here.
>
> [1] Liang Yao, Chengsheng Mao, and Yuan Luo. Kg-bert: Bert for knowledge graph completion. arXiv preprint arXiv:1909.03193, 2019.
>
> [2] Xiaozhi Wang, Tianyu Gao, Zhaocheng Zhu, Zhengyan Zhang, Zhiyuan Liu, Juanzi Li, and Jian Tang. Kepler: A unified model for knowledge embedding and pre-trained language representation. Transactions of the Association for Computational Linguistics, 9:176–194, 2021.
>
> [3] Kalev Leetaru and Philip A Schrodt. Gdelt: Global data on events, location, and tone, 1979–2012. In ISA annual convention, volume 2, pp. 1–49. Citeseer, 2013.
>
> **Question 2**: Based on the results we can see that the ECOLA mainly leverages the language model to improve the KG embeddings. I wonder if the performance can be further improved if larger language models are leveraged since BERT base is relatively small.
>
> **Reply**: Thank you very much for your suggestion. We are running experiments with larger language models instead of Bert-base. We will update the results in a later comment.
>
> **Question 3**: Does the quality of the text significantly affect the model’s performance? If in another domain there is no high-quality paired text, can the proposed method still improve the KG embeddings?
>
> **Reply**: Yes, the quality of the text does affect the model’s performance.  Taking the DuEE dataset as an example, the alignment between text and quadruples is not so satisfying. The entities in the corresponding knowledge graph are often broad concepts, such as New Energy Vehicle Company. But most texts are specific to subconcepts of the entity, such as Tesla, NIO, etc. Thus, the alignment between texts and entities is worse than on the GDELT dataset. Table 1 in the manuscript also shows that the performance gain on DuEE is smaller compared to the GDELT dataset.

---

> > ### Author Response · Authors · 2022-11-14
> > **Response to Reviewer cTPN, part B**
> >
> > **Reply to Weakness 1 regarding the Model Variants**
> >
> > The model variants shown in the manuscript are intended to show that our model is model-agnostic and can generally boost various temporal knowledge graph models, and thus, our work should have a considerable impact on the community.
> >
> > However, the model variants in our work are not limited to combinations of ECOLA with different previous knowledge embedding methods. We have tried different strategies to incorporate textual knowledge into temporal knowledge graph embeddings. For example, we tried to use texts to boost the static part of entity embeddings only, which corresponds to **ECOLA-SF** mentioned in the paragraph of *Quantitative Study* and *Appendix B*.  ECOLA-SF is the static counterpart of the proposed approach ECOLA, where we do not consider the temporal alignment between time-dependent entity representation and textual knowledge. As shown in Figure 4a, ECOLA-SF is empirically outperformed by ECOLA, demonstrating that temporal alignment is more powerful than static alignment.
> >
> > Besides, we also explored ways to incorporate the knowledge embeddings enhanced by the textual information and the knowledge embeddings learned by the temporal KG score function (tKG loss). For example, we tried a post-integration approach, where we learn two tables of entity embedding by KTP loss and tKG loss respectively, and then fuse them later with various aggregation methods. Such model variants perform worse than ECOLA, such that we only have ECOLA in the final manuscript.
> >
> > **Reply to Weakness 2 regarding the Language Model Gain after the Joint Training**
> >
> > Yes, we selected temporal question answering as a downstream task to show that the proposed ECOLA can benefit the language model as well. Natural questions often include temporal constraints, e.g., who was the US president *before Jimmy Carter?* To deal with such challenging temporal constraints, temporal question answering over tKGs has become trendy since tKGs help to find entity or timestamp answers with support of temporal facts. Saxena et al. [4] introduced the tKBQA dataset *CronQuestions* containing natural temporal questions with different types of temporal constraints and an accompanying temporal knowledge graph (tKG). They proposed a baseline called *CronKGQA* that uses a pre-trained language model (BERT) to understand the natural questions followed by a scoring function for answer prediction.  We enhance the language encoder in *CronKGQA with* our proposed ECOLA approach, i.e., we find temporal relevant texts for quadruples in the supporting tKG in CronQuestions and train the language model and the tKG model jointly with our proposed KTP task. Then we plug the enhanced language model back into *CronKGQA*. Empirical results show that our proposed ECOLA enhances the language model with **7.4%** relative improvements regarding *precision* on CronQuestions, demonstrating the benefits to the language model.
> >
> > [4] Saxena, A.; Chakrabarti, S.; and Talukdar, P. 2021. Question Answering over Temporal Knowledge Graphs. In ACL. Saxena, A.; Tripathi, A.; and Talukdar, P. 2020. Improving Multi-hop Question Answering over Knowledge Graphs using Knowledge Base Embeddings. In ACL.

---

### Official Review · Reviewer_LrUQ · 2022-10-27

**Confidence:** 4
**Correctness:** 2
**Technical Novelty And Significance:** 2
**Empirical Novelty And Significance:** 3
**Recommendation:** 5

**Clarity, Quality, Novelty And Reproducibility:**

Question: as mentioned in the section 3 introduction, no textual data is used during inference time, does it mean that the knowledge triples still need to pass through the transformer encoder? Or only the tKG model is used during inference.


In terms of novelty, incorporating contextualized embedding into the temporal knowledge graph completion task has been proposed. The main modification of ECOLA is to use an additional knowledge-text prediction task to jointly optimize the pre-trained language models. However, it is also possible that the performance gain is due to the encoder’s capability from the pre-trained language model. It would be beneficial if the authors can show the experiments without KTP loss and give more insights into how the KTP task benefits the temporal knowledge graph completion task.


**Strength And Weaknesses:**

Strength: This paper proposed a technique to enhance the temporal knowledge graph embeddings by contextualized language models. The motivation is reasonable and experiment results have demonstrated decent improvements for the temporal KG completion tasks.

Weakness:
1. Enhancing knowledge graph embeddings by contextualized language model embedding itself has been widely explored, even for enhancing temporal knowledge graph embedding [1].

2. It is unclear how the transformer encoder (BERT) combines with the tKG model in Figure 2. From section 3.1, we know the quadruple representation can be obtained by the BERT encoder. However, the introduction of DyERNIE is confusing. Since the embedding of DyERNIE is derived from a static embedding and a velocity vector, how does this equation link to the notations in section 3.1? The title of section 3.1 is ‘Embedding layer’, whereas the final representation of BERT should be dynamic contextualized embedding since the BERT encoder is updated by the KTP (MLM-like) loss. In figure 3, the time expression 2019.08.02 is an input to the transformer, but it seems to be treated differently compared to the other words. The notation tau mentioned in figure 3 was not explained elsewhere, whereas tau is related to the most important temporal aspect of this model.


[1] Mavromatis, Costas, et al. "Tempoqr: temporal question reasoning over knowledge graphs." Proceedings of AAAI. 2022.



**Summary Of The Paper:**

This paper proposed to use contextualized language representation to enhance the temporal knowledge graph embedding. It combines tKGE losses with knowledge-text-prediction loss during training. As such, the proposed method can be combined with arbitrary tKGE methods. Experimental results showed significant improvements in various temporal knowledge graph completion tasks.

**Summary Of The Review:**

In summary, this paper proposed a framework that enhances the temporal knowledge graph embedding with pre-trained language models and jointly optimizes the TKGC task with knowledge-text prediction loss (KTP). The proposed model has demonstrated decent experimental improvements in the temporal knowledge graph completion tasks.  However, this paper lacks sufficient details on the model structure. It is not clear how the pre-trained transformer model combines with tKG model.


Besides, the analysis section of the paper can also be improved. For example, in the qualitative analysis part, the examples only showed that ECOLA-DE benefited from the extra relevant textual training data. Whereas this finding is somewhat superficial, and it is not directly related to the proposed KTP loss. Further analysis can be conducted on the underlying reasons behind ECOLA’s strong empirical performance.

==============================
Updates on Nov 30.

I appreciate the clarification of the authors, which help me understand some parts better. The authors also mentioned that they will update their paper accordingly. Therefore, I made an adjustment to my recommendation score. Overall, the paper's technique contribution is not significant enough, though the proposed method for combining the two sources improved the performance.

---

> ### Author Response · Authors · 2022-11-07
> **Response to Reviewer LrUQ, Part A**
>
> **Thank you very much! We appreciate your effort in reviewing our paper.**
>
> **But we believe that you have a considerable misunderstanding regarding our work as well as the task of temporal knowledge graph completion.  We will address your concerns and questions in the following. Please be sure to read this before you make the final judgment. If you have any other questions, please let us know. I'm looking forward to your reply.**
>
> **Your statement of Weakness 1**:  *Enhancing knowledge graph embeddings by contextualized language model embedding itself has been widely explored, even for enhancing temporal knowledge graph embedding [1].*
>
> [1] Mavromatis, Costas, et al. "Tempoqr: temporal question reasoning over knowledge graphs." Proceedings of AAAI. 2022.
>
> **Reply**: We believe [1] did not explore enhancing temporal knowledge graph embedding by contextualized language model embedding. And this field has not been well explored. First of all, TempoQR [1] is a framework to answer natural language questions using temporal knowledge graphs. They **do not** address the temporal knowledge graph completion task and also **do not** enhance temporal knowledge graph embedding. As mentioned in Section 5.1 in [1],  *the pre-trained language model parameters and the TKG embeddings **are not updated***. In other words, TempoQR leverages pre-trained TKG embeddings to answer complex natural language questions, but they **do not** focus on enhancing TKG embedding. Secondly, [1] only evaluated TempoQR on the temporal Question Answering task. They used a benchmark dataset called CronQuestions and reported the results on this dataset. However, TemporalQR is not evaluated on the temporal knowledge graph completion (tKGC) task. The tKGC task is a kind of link prediction task. Let $\mathcal W$ represent the set of all temporal facts (quadruples) of the tKGC task, and $\mathcal G$ is a subset of $\mathcal W$, e.g., the incomplete TKG. TKGC is the problem of inferring $\mathcal W$ from $\mathcal G$. More details of the definition of temporal knowledge graph completion task can be found in [2,3].
>
> [2] Goel, R., Kazemi, S. M., Brubaker, M., & Poupart, P. (2020, April). Diachronic embedding for temporal knowledge graph completion. In Proceedings of the AAAI Conference on Artificial Intelligence.
>
> [3] Zhen Han, Gengyuan Zhang, Yunpu Ma, and Volker Tresp. 2021. Time-dependent Entity Embedding is not All You Need: A Re-evaluation of Temporal Knowledge Graph Completion Models under a Unified Framework. In Proceedings of the 2021 Conference on Empirical Methods in Natural Language Processing.

---

> > ### Author Response · Authors · 2022-11-07
> > **Response to Reviewer LrUQ, Part B**
> >
> > **Your statements of weakness 2**:
> >
> >  *1. It is unclear how the transformer encoder (BERT) combines with the tKG model in Figure 2.*
> >
> >   **Reply:** Thank you for pointing out our imperfections in Figure 2, and we hope this explanation finds you well. A short answer is that the transformer encoder (BERT) and the tKG model share **the same knowledge embedding layer**, i.e., entity lookup embeddings and predicate lookup embeddings.
> >
> > Figure 2 of our model structure is suggested to be read in bottom-up order.  An input sample of the transformer encoder has two parts concatenated, i.e.,  *“South Korea is moving Japan from a list of trusted trading partners on Aug. 12, escalating a dispute with its neighbor.” followed by “(South Korea, downgrades trade ties with, Japan, 2019.08.02)”*
> >
> > The first part “*South Korea is moving Japan from a list of trusted trading partners on Aug. 12, escalating a dispute with its neighbor.* ” is the aligned textual description extracted from news resources of the second part *(South Korea, downgrades trade ties with, Japan, 2019.08.02)*. These two parts are encoded differently before feeding to the transformer to do the KTP task, as described in Section 3.1 in our manuscript. The first part will be encoded by the WordPiece embeddings, and the second part will be encoded by entities and predicates lookup tables to get knowledge embeddings. Then these two kinds of embeddings will be concatenated with a [SEP] token and masked with different strategies in Section 3.4 and get the form *“[CLS] South Korea is moving Japan from <mask> trusted trading partners on Aug. 12, escalating a dispute with its neighbor. [SEP] (<entity mask>, <predicate mask>, Japan, 2019.08.02)”* to feed the transformer encoder for the KTP task.
> >
> > In other words, each of the investigated tKG models in our paper maintains lookup tables for entities and predicates, which are also used to generate the initial embeddings of the second part of the input to the transformer encoder. As you may see, there are two arrows stemming from the right bottom tKG model in Figure 2. This represents that entities and predicates in the tKG and KTP part of our model share the same embedding lookup table during training.
> >
> >
> > *2. (**Your Statement**) From section 3.1, we know the quadruple representation can be obtained by the BERT encoder.  However, the introduction of DyERNIE is confusing. Since the embedding of DyERNIE is derived from a static embedding and a velocity vector, how does this equation link to the notations in section 3.1?*
> >
> > **Reply:** As explained in the previous paragraph, the quadruple representation is not obtained by the BERT but by the lookup table in the tKG model and is then fed into the BERT encoder together with its textual description of the quadruple.  Therefore, the time-dependent entity embedding of an entity $e_i^{DyER(t)}$ derived from the DyERNIE model should be linked to $e_i(t)$ in Section 3.1 with the superscript DyER indicating that this entity representation is obtained from the DyERNIE model. Graphically, $e_i^{DyER(t)}$ should take the position of $E(\tau)$ in Figure 3 where the $\tau$ here represents time $t$. Thanks for pointing this out and we have revised our paper regarding this time notation.
> >
> > *3. (**Your Statement**) The title of section 3.1 is ‘Embedding layer’, whereas the final representation of BERT should be dynamic contextualized embedding since the BERT encoder is updated by the KTP (MLM-like) loss.*
> >
> > **Reply:** In Section 3.1 we only talked about how we designed the input embedding layer before we fed it into the BERT encoder. Thus, Section 3.1 is only about Figure 3.  The token embeddings of the input contain word embeddings followed by knowledge embeddings. Then the type embeddings and position embeddings differentiating tokens between the text and knowledge parts are added on top of token embeddings. The contextualized embeddings you mentioned are introduced later in Section 3.3 and Section 3.4.

---

> > > ### Author Response · Authors · 2022-11-07
> > > **Response to Reviewer LrUQ, Part C**
> > >
> > > **Continued with your statements of weakness 2**:
> > >
> > > *4. (**Your Statement**) In figure 3, the time expression 2019.08.02 is an input to the transformer, but it seems to be treated differently compared to the other words.*
> > >
> > > **Reply:** The example time expression 2019-08-02 in Figure 3 is not a word token. In the input *“South Korea is moving Japan from a list of trusted trading partners on Aug. 12, escalating a dispute with its neighbor. (South Korea, downgrades trade ties with, Japan, 2019-08-02)”*, following the textual tokens, the last four tokens correspond to a quadruple from the temporal knowledge graph. So ”2019-08-02” is a timestamp in the quadruple (South Korea, downgrades trade ties with, Japan, 2019-08-12) providing temporal information in the tKG model and will be taken as $t$ in the entity representation $e_i(t)$. Therefore it is treated differently compared to other word tokens where each word has an embedding. Here the timestamp 2019-08-02 doesn’t have an embedding since the investigated tKG models don’t have a lookup table for time but incorporate the temporal information in entity embeddings.
> > >
> > > *5. (**Your Statement**) The notation tau mentioned in figure 3 was not explained elsewhere, whereas tau is related to the most important temporal aspect of this model.*
> > >
> > > **Reply:** Thanks for pointing this out and we have revised our paper regarding this time notation. As replied in Argument 2, the $\tau$ is equal to the time information $t$ in the quadruple for obtaining the temporal embedding $e_i(t)$ for an entity. We have replaced $\tau$ with $t$ in the updated version.

---

> > > > ### Author Response · Authors · 2022-11-07
> > > > **Response to Reviewer LrUQ, Part D**
> > > >
> > > > **Your comments** regarding **Clarity, Quality, Novelty, And Reproducibility:**
> > > >
> > > > 1. *Question: as mentioned in the section 3 introduction, no textual data is used during inference time, does it mean that the knowledge triples still need to pass through the transformer encoder?* *Or only the tKG model is used during inference.*
> > > >
> > > >
> > > >      **Reply**:
> > > >       No, the knowledge triples **do not** pass through the transformer encoder. Only the tKG model is used during the inference.
> > > >
> > > >
> > > > 2. *In terms of novelty, incorporating contextualized embedding into the temporal knowledge graph completion task has been proposed.*
> > > >
> > > >     **Reply**:  I was wondering which works were proposed to incorporate contextualized embedding into the temporal knowledge graph completion task. Can you please point them out?
> > > >
> > > >     Besides, the objective of our paper is not only to incorporate conceptualized embedding into the temporal knowledge graph completion task but also to examine how to homogenize the structured knowledge from TKG and the unstructured textual knowledge, for example, if we have relevant textual data about a TKG, how to inject textual information into **temporal**
> > > >     knowledge embedding. A challenge here is that the entities in TKG often have dynamic representations. Therefore, given an entity, it should be taken into account which textual knowledge is relevant to it at which **timestamp**. We addressed this challenge in our paper.
> > > >
> > > > 3. *The main modification of ECOLA is to use an additional knowledge-text prediction task to jointly optimize the pre-trained language models. However, it is also possible that the performance gain is due to the encoder’s capability from the pre-trained language model. It would be beneficial if the authors can show the experiments without KTP loss and give more insights into how the KTP task benefits the temporal knowledge graph completion task.*
> > > >
> > > >       **Reply**: As mentioned above, we **do not** use textual data during inference time, and also the knowledge triples **do not** need to pass through the transformer encoder. In other words, we only use the entity embedding and predicate embedding from the tKG model to predict links at the inference time. Thus, the performance gain at inference time is not due to the encoder’s capability from the pre-trained language model but because of the enhanced knowledge embedding. **Note that** in our paper, entity embedding and predicate embedding denote the knowledge embedding from the lookup tables, i.e., entity embedding table and predicate embedding table, instead of the representations after the BERT encoder. In our model, the KTP task and the tKG model share the same embedding layer of entities and predicates. By optimizing the entity embedding and predicate embedding using both the KTP loss and the tKG loss, the entity embedding and predicted embedding got enhanced.
> > > >
> > > >     Therefore, as shown in Table 1 in the manuscript, the performance difference between **UTEE** (a TKG model) and **ECOLA-UTEE** (our model) as well as the performance gap between **DyERNIE** (another TKG model) **and ECOLA-DyERNIE** (our model) clearly show the benefits of the KTP task. UTEE and DyERNIE are exactly the experiments **without** the KTP loss.  In other words, ECOLA-UTEE downgrades to UTEE by removing the KTP loss. To put it more clearly,  the difference between UTEE and ECOLA-UTEE is that the knowledge embedding in ECOLA-UTEE is jointly optimized by the KTP loss and the UTEE (tKG) loss, but the knowledge embedding in ECOLA is only optimized by the UTEE (tKG) loss. The procedure at inference time is the **same** for both UTEE and ECOLA-UTEE: we get temporal knowledge embeddings from the lookup tables and use the score function of UTEE to predict the missing links. Please refer to Quantitative Study and Qualitative Analysis in Section 6 of our manuscript for insights of the KTP task.

---

> ### Comment · Reviewer_LrUQ · 2022-11-15
> **In response to the authors response until Nov 8**
>
> Even though CronQuestions is a TKG QA task, it is closely related to the TKGC task. The formulation of the questions in CronQuestions is also similar to link predictions in temporal KG. The discussion can be found in 'Background' in [1].
>
> On reference [1], this work combines the pre-trained language model (PrLMs) with tKG embedding model TComplEx. Whereas ECOLA showed that it combines BERT with DyERNIE etc. These two works made use of PrLM together with tKG embedding models. Even though the TEmpoqr model did not update the tKG embeddings, it used a new transformer encoder and the self-attention mechanism to learn the interactions between tKG embeddings and textual input. Besides, many prior works leverage PrLM with conventional KG embeddings as well, such as KEPLER.
>
> I agree with #Reviewer REq6's comments that this paper basically extends the conventional KG+LM formulation to tKG + LMs setting. The KTP loss and leveraging the contextualized representations are not tackling the temporal factor of the TKGC task. This factor is mainly handled by the TKE model. Finding temporal relevant textual data for temporal knowledge quadruples has a trivial contribution. A similar formulation of KEPLER could also be combined with TKE models such as DyERNIE.
>
> To me, the main contributions of this paper are (1) combining the LM encoder with tKG models and (2) Jointly optimizing tKG model with MLM-like KTP loss. The main weakness of this paper is the unclear description of its main contributions.
>
> In the rebuttal, the authors replied that only the entity embedding and predicate embedding layers are shared with PrLM. That is to say, the 'Embedding' row of Figure 3 actually has three different embedding lookup tables with different vocabulary sizes (i.e. workpiece vocabulary, entity vocabulary, predicate vocabulary). That is to say, these inputs are from different vector spaces. There are two E[Japan]s in Figure 3 and they are from two different embedding spaces. There are no mathematical formulations that describe the conversion from different types of inputs to different types of embeddings, however, this seems to be the most important part of the paper. Figure 3 showed an illustration of embedding summation that is almost identical to BERT [2], whereas figure 3 did not show the differences between 'knowledge tokens' and normal tokens.
>
> By the convention of masked language modeling, the final contextualized representations (after the last layer of PrLM) are used for token prediction. However, the authors clarified that only the combined embeddings (after layer 1 of PrLM) are fed into the tKG models. This part is also not clearly described in the paper. Since most works leverage the final contextualized representation for downstream tasks, why would you only use the embedding layer for the tKG models? Intuitively, the contextualized representation at the last layer can also be used to score the possible answers. Since the transformer encoder has been trained by the knowledge-intensive KTP task, using the final contextualized representation should at least be a baseline for ECOLA. In my opinion, finding a model architecture that performs empirically well is insufficient for acceptance by top conferences such as ICLR.
>
>
> [1]. Mavromatis, et al. "Tempoqr: temporal question reasoning over knowledge graphs." Proceedings of AAAI. 2022.
>
> [2]. Wang,  et al. "KEPLER: A unified model for knowledge embedding and pre-trained language representation." Transactions of ACL, 2021
>
> [3]. Devlin et al,"BERT: Pre-training of Deep Bidirectional Transformers for Language Understanding", Proceedings of NAACL, 2019.

---

> > ### Author Response · Authors · 2022-11-18
> > **Response to Reviewer LrUQ's response on Nov 15**
> >
> > **Reply:**
> >
> > **Thank you very much for reading our response and giving more suggestions. We appreciate your feedback.**
> >
> > According to your suggestions, we revised the Figure 3 as well as Section 3.1 *Embedding Layer* in our manuscript to emphasize the differences between *knowledge tokens* and *normal subword tokens*.
> >
> > **Regarding your question** "*Since most works leverage the final contextualized representation for downstream tasks, why would you only use the embedding layer for the tKG models?*", the reasons are two folds. **Firstly**, we usually don’t have textual data at inference time of the temporal knowledge graph completion (tKGC) task. We starts from this point of view and define the experimental settings such that the model only use the embedding layer for the tKG models. **Secondly**, only using the embedding layer at inference time makes the comparison of ECOLA with other baseline tKG embedding models more intuitively, and thus, demonstrates how the KTP task benefits the tKGC task. If we use the contextualized representation at the last layer, then the performance gain could also be due to the encoder’s capability from the pre-trained language model. **Nevertheless**, we agree that using the final contextualized representation could be as a baseline for ECOLA. We will show the results in the final version.
> >
> > Finally, we would like to emphasize that KEPLER is not able to deal with the temporal relevant textual data well. Empirical results show that our proposed ECOLA far outperforms a similar formulation of KEPLER combined with TKE models .

---

### Official Review · Reviewer_REq6 · 2022-10-30

**Confidence:** 4
**Correctness:** 4
**Technical Novelty And Significance:** 2
**Empirical Novelty And Significance:** 3
**Recommendation:** 5

**Clarity, Quality, Novelty And Reproducibility:**

The paper is well-written and easy to understand. As mentioned above, I think it lacks novelty. Code and data are included with the submission.

A question: What are b_i and b_j in (1)? I did not find the definition in the paper.

**Strength And Weaknesses:**

Strength:

The paper fills in the gap on using temporal KG to enhance the language model. The results look solid and bring consistent improvement in link prediction.

Weakness:
1. The method in the paper lacks novelty. It basically reformulates the previous KG+LM methods to the temporal KG+LM setting. Actually, the method (and even the figures) looks a lot like the KEPLER paper (Wang et al., 2019), with the KG embeddings replaced by tKG embeddings, and the transE loss replaced by the corresponding tKG loss.
2. The method is only tested on link prediction tasks. I would expect this model to work in more diverse settings (i.e., any task that LM can be used for). It would be great if the model could work for settings like question answering, event extraction, etc. I'm also interested in how the tKG-pretraining affects general NLU performance like on GLUE.

**Summary Of The Paper:**

The paper proposes a temporal KG-enhanced language model.

The idea is to combine pretrained temporal knowledge embeddings with the language model as input tokens, and the language model will take temporal-aware embeddings as input.
The authors propose masking the predicates, entities, or subwords for pretraining the model.
The authors convert three KGs into the pretraining corpus to find relevant text with temporal KG knowledge. Experiments show that the pretrained model has improved performance on link prediction for the three temporal KGs.

**Summary Of The Review:**

In all, I feel the paper focuses on an important area, but the novelty and contribution are not enough for a conference like ICLR.

---

> ### Author Response · Authors · 2022-11-08
> **Response to Reviewer REq6, Part A**
>
> **Thank you very much for your helpful feedback and valuable comments! We try to resolve the issues and incorporate the comments in our updated draft.**
>
> *Weakness 1: The method in the paper lacks novelty. It basically reformulates the previous KG+LM methods to the temporal KG+LM setting. Actually, the method (and even the figures) looks a lot like the KEPLER paper (Wang et al., 2019), with the KG embeddings replaced by tKG embeddings, and the transE loss replaced by the corresponding tKG loss.*
>
> **Reply:**
>
> [**Limitations of KEPLER**]
> First of all, we must point out that KEPLER cannot be applied to tKG models since it lacks the ability to catch the temporal dynamics of evolving knowledge in tKG.  Specifically, the entity embedding of KEPLER is obtained by encoding its textual descriptions with a language model. In KEPLER, each entity, no matter at which timestamp, shares the same static embedding generated from a shared entity description. However, entity embedding in tKG models is often time-evolving, i.e., parametrized by a function of time. This is why KEPLER cannot handle tKG models due to its limitation of assuming knowledge embedding is static and using the time-invariant description of an entity to generate its representation. In other words, a time-invariant entity description is not eligible for generating time-involving entity embeddings. If KEPLER is applied to tKG directly, it loses dynamic information and downgrades the tKG embeddings to static KG embeddings. We have done some experiments with adapted KEPLER to tKG models, but there’s no performance gain.
>
> [**How our model differentiates from KEPLER**]
> Therefore, to address this problem, we proposed ECOLA which shows superior performance gain with two main contributions: 1) solving the temporal alignment challenge between tKG and temporal relevant textual data; 2) injecting the temporal relevant textual knowledge into the temporal knowledge graph embedding through the KTP task.
>
> Regarding Contribution 1), given an entity in tKG, it should be taken into account which textual knowledge is relevant to it at which timestamp. Therefore, it is essential to solve the *temporal alignment challenge* between texts and tKG, which is to establish a correspondence between textual knowledge and their temporal knowledge graph depiction. In the implementation, We pair a **quadruple** with its temporal-relevant textual data, e.g., event descriptions, which corresponds to the temporal relations between entities at a specific time.
>
> Regarding Contribution 2), to inject the relevant textual knowledge into the temporal knowledge graph embedding, we proposed the KTP task consisting of three parts, i.e. word prediction, relation prediction, and entity prediction. As explained in Section 3.4, the task of predicting masked entities in the KTP task enables ECOLA to learn entity embedding through both unmasked word tokens and knowledge tokens, which can integrate unstructured temporal textual knowledge and structured knowledge from tKG.
>
> Even though there are two losses in both KEPLER and our ECOLA setting, their functionalities are quite different. The goal of the MLM loss in KEPLER is only to avoid catastrophic forgetting of the language model during training the KE objective. And thus, the transformer encoder of KEPLER only requires unstructured texts. In comparison, the KTP loss (an extended MLM loss) in our model is the key for aligning the language representations and temporal knowledge embeddings. It requires both unstructured text and temporal knowledge graphs quadruple and integrates the unstructured knowledge and unstructured knowledge by predicting the masked entities and predicates.
>
> [**Efficiency comparison**]
> What's more, KEPLER uses very large textual corpora for the MLM objective, i.e., EnglishWikipedia(13GB), to maintain the general language understanding ability. It has much worse performance if finetuned on a smaller corpus formed from only entity descriptions. In comparison, our ECOLA solves the temporal alignment between tKG and text, and manages to use quite small datasets (577M vs. 13GB) formed from paired event descriptions and the aligned knowledge quadruples. Still, we have significant performance gains across different tKG models.

---

> > ### Author Response · Authors · 2022-11-08
> > **Response to Reviewer REq6, Part B**
> >
> > *Weakness 2: The method is only tested on link prediction tasks. I would expect this model to work in more diverse settings (i.e., any task that LM can be used for). It would be great if the model could work for settings like question answering, event extraction, etc. I'm also interested in how the tKG-pretraining affects general NLU performance like on GLUE.*
> >
> > **Reply:** While most works focus on using knowledge embeddings to enhance language models, our work studied how to enhance temporal knowledge graph embedding using textual knowledge and pre-trained language representations and to improve the link prediction task on temporal knowledge graphs. This problem is quite challenging since the entity embeddings in tKG models could change over time, which means it should be taken into account which textual knowledge is temporally relevant to an entity embedding at which timestamp.
> >
> > Regarding other NLP tasks that enhanced LM can be used for,  we are conducting experiments on the temporal question-answering task, i.e., the CronQuestions dataset, to evaluate the language model enhanced by temporal knowledge embeddings. Since it’s out of the scope of this paper (as the title of this paper shows), we would like to finish it in future work.
> >
> > *Question in Clarity, Quality, Novelty And Reproducibility: What are b_i and b_j in (1)? I did not find the definition in the paper.*
> >
> >
> > **Reply:** Thank you very much for pointing it out. We have updated our manuscript and added an explanation. Specifically, $b_i, b_j \in \mathbb R$ are entity-specific scalar biases according to entity $e_i$ and $e_j$ respectively.

---

> ### Author Response · Authors · 2022-11-18
> **Response to Reviewer REq6, Update on Nov. 18**
>
> Dear Reviewer REq6,
>
> **We would like to update you regarding the second weakness you mentioned**, which is "*The method is only tested on link prediction tasks. I would expect this model to work in more diverse settings (i.e., any task that LM can be used for). It would be great if the model could work for settings like question answering, event extraction, etc. I'm also interested in how the tKG-pretraining affects general NLU performance like on GLUE.*"
>
> To address your interest in our model’s enhancement of language model in the NLU task, we selected **temporal question answering** as a downstream task to show that the proposed ECOLA can benefit the language model as well. We will show you a summary of the experiments in the following. For more details please refer to Appendix C.
>
> Natural questions often include temporal constraints, e.g., who was the US president before Jimmy Carter? To deal with such challenging temporal constraints, temporal question answering over tKGs has become trendy since tKGs help to find entity or timestamp answers with support of temporal facts. Saxena et al. [4] introduced the tKBQA dataset CronQuestions containing natural temporal questions with different types of temporal constraints and an accompanying temporal knowledge graph (tKG). They proposed a baseline called CronKGQA that uses a pre-trained language model (BERT) to understand the natural questions followed by a scoring function for answer prediction.  We enhance the language encoder in CronKGQA with our proposed ECOLA approach, i.e., we find temporal relevant texts for quadruples in the supporting tKG in CronQuestions and train the language model and the tKG model jointly with our proposed KTP task. Then we plug the enhanced language model back into CronKGQA. Empirical results show that our proposed ECOLA enhances the language model with 7.4% relative improvements regarding precision on CronQuestions, demonstrating the benefits to the language model.

---

### Decision · Program_Chairs · 2023-01-20

**Decision:**

Reject

**Justification For Why Not Higher Score:**

This is clearly a borderline paper that does not have strong supports nor oppositions. While the paper's motivation and experimental results are solid, as most reviewers are concerned, the novelty of the approach is rather limited; the paper can be viewed as making changes to existing methods that can improve their performance in the given task. While these achievements should not be undervalued, considering the characteristics of ICLR and its audience, the paper's weaknesses seem to outweigh its strengths in this venue and might not have enough interested audience.

**Justification For Why Not Lower Score:**

N/A

**Metareview: Summary, Strengths And Weaknesses:**

Summary: This paper presents a method for improving temporal knowledge graph embedding by using pre-trained language models and optimizing the TKGC task with a knowledge-text prediction loss. The proposed approach showed promising results in temporal knowledge graph completion tasks.

Strengths:
- All reviewers agree that the motivation of the method and the experimental results (improvements) are convincing.
- All reviewers agree that the paper is clearly written. Unclear parts were clarified after rebuttals.

Weaknesses:
- Most reviewers are concerned with the novelty; that is, the proposed method is not too different from previous methods (e.g. KG+LM).